# Magnetic Iron Oxide Nanoparticles Coated by Coumarin-Bound Copolymer for Enhanced Magneto- and Photothermal Heating and Luminescent Thermometry

**DOI:** 10.3390/nano14110906

**Published:** 2024-05-22

**Authors:** Alexiane Féron, Sylvain Catrouillet, Saad Sene, Gautier Félix, Belkacem Tarek Benkhaled, Vincent Lapinte, Yannick Guari, Joulia Larionova

**Affiliations:** ICGM, Univ Montpellier, CNRS, ENSCM, Montpellier, France; alexiane.feron@etu.umontpellier.fr (A.F.); saad.sene@umontpellier.fr (S.S.); gautier.felix@umontpellier.fr (G.F.); belkacem-tarek.benkhaled@umontpellier.fr (B.T.B.); vincent.lapinte@umontpellier.fr (V.L.)

**Keywords:** nanoparticles, polymeric coating, luminescence, thermometry, coumarin

## Abstract

In this work, we report on the synthesis and investigation of new hybrid multifunctional iron oxide nanoparticles (IONPs) coated by coumarin-bound copolymer, which combine magneto- or photothermal heating with luminescent thermometry. A series of amphiphilic block copolymers, including Coum-C_11_-PPhOx_27_-PMOx_59_ and Coum-C_11_-PButOx_8_-PMOx_42_ bearing luminescent and photodimerizable coumarin moiety, as well as coumarin-free PPhOx_27_-PMOx_57_, were evaluated for their utility as luminescent thermometers and for encapsulating spherical 26 nm IONPs. The obtained IONP@Coum-C_11_-PPhOx_27_-PMOx_59_ nano-objects are perfectly dispersible in water and able to provide macroscopic heating remotely triggered by an alternating current magnetic field (AMF) with a specific absorption rate (SAR) value of 240 W.g^−1^ or laser irradiation with a photothermal conversion efficiency of η = 68%. On the other hand, they exhibit temperature-dependent emission of coumarin offering the function of luminescent thermometer, which operates in the visible region between 20 °C and 60 °C in water displaying a maximal relative thermal sensitivity (S_r_) of 1.53%·°C^−1^ at 60 °C.

## 1. Introduction

Inorganic nanoparticles capable of inducing a macroscopic temperature increase when exposed to external *stimuli*, such as alternating current magnetic fields (AMFs) or light irradiation, have garnered significant attention for several decades. This attention stems not only from a fundamental standpoint but also due to their promising applications for hyperthermia therapy for cancer [1,2,3], bacterial infections [4,5,6,7,8], drug delivery systems [9], modulation of enzymatic reactions [10], plasmonic devices, control of single cell functions [11,12], catalytic processes [13], polymerization reactions [14] and more. A plethora of nanostructures have been engineered to serve as photothermal or magnetothermal nano-heaters, encompassing plasmonic metallic nanostructures, diverse carbon materials, metal oxides, metal alloys, semiconductors and carbides/nitrides, each offering distinct advantages for targeted heating applications [15,16,17,18]. Among these, iron oxide nanoparticles (IONPs) have been the subject of extensive investigation for many decades, owing to their exceptional capacity to generate heat when remotely exposed to external *stimuli*, their controlled size ranging from a few to hundred nanometres, different shapes, an easily functionalizable surface and biocompatibility [18,19,20]. Indeed, they can efficiently convert magnetic energy into heat providing an important temperature rise of colloidal solutions containing nanoparticles at macroscopic level. Recent advancements have led to the development of highly efficient IONPs with various shapes, such as spherical, cubic or dendritic forms, achieving specific absorption rate (SAR) values of up to 1000 W g^−1^ [18,21]. More recently, IONPs have also been recognized as photothermal agents capable of providing an important temperature rise under irradiation in the near-infrared (NIR) window, making them competitive with conventional Au nano-objects [21]. Moreover, the possibility of employing them as dual magneto- and photothermal agents, therefore enhancing the heating ability by up to fivefold when compared to magnetic stimulation alone, reaching SAR values of up to 5000 W g^−1^, has also been demonstrated [22].

One of the primary challenges in the realm of nanoparticle-assisted heating lies in achieving precise temperature control at the nanoparticle’s surface. This task is especially critical due to the rapid fluctuations and localized effects inherent at such scales, necessitating innovative strategies for accurate temperature monitoring and regulation. This issue holds principal importance for optimizing various applications, including hyperthermia therapy, catalytic processes and polymerization reactions, all of which rely on maintaining specific temperature ranges or a very localized heating for optimal performance. However, conventional temperature measurement techniques, such as thermocouples or infrared thermography, often struggle with limitations in sensitivity and accuracy when applied to nanoscale systems. Fluorescence-based thermometry is emerging as one of the most promising alternative approaches, aimed at overcoming these challenges and enabling more precise local and remote temperature monitoring at the micro- and even nanoscales [23,24,25,26]. In this context, the design of multifunctional heater/luminescent thermometer nano-objects is essential for ensuring nanoparticle-assisted heating triggered by external *stimuli*, which is combined with accurate luminescent-based temperature monitoring at the nanoscale. Such design requires a careful choice of nano-heater, including its size, composition and surface functionalization, as well as the selection of appropriate luminescent probes and identification of their interface. In this line of thought, three main approaches have been implemented with IONPs. Firstly, L. D. Carlos, A. Millan and colleagues reported on the successful combination of magnetic γ-Fe_2_O_3_ nanoparticles with Ln^3+^-based β-diketonate coordination complexes (where Ln^3+^ = Tb^3+^/Eu^3+^, Tb^3+^/Sm^3+^) as ratiometric luminescent temperature probes based on the luminescent intensity ratio of two lanthanides included into the organosilica [27] or copolymer shells [28,29]. In these works, the luminescent complexes have been encapsulated into the shells without their covalent anchoring. Secondly, both upconversion NaYF_4_:Yb^3+^, Er^3+^ nanorods permitting temperature measurements based on their temperature-dependent lifetime and small IONPs allowing magnetothermal heating have been included in mesoporous silica nanoparticles [30]. This work demonstrated that the temperature inside the silica nano-objects containing magnetically activated IONPs was higher in comparison with the macroscopic ones in environmental colloidal solution. Thirdly, magnetic nanoparticles of several compositions and sizes (spherical MnFe_3_O_4_ of 6 nm, MnCoFeO_4_ of 15 nm, spherical Fe_3_O_4_ nanoparticles of 12 or truncated octahedral of 25 nm) have been conjugated with organic dyes (Rhodamine B or DyLight) [12,31,32] or with fluorescent proteins, which present temperature-dependent fluorescence permitting the temperature readout [10]. Note that for these latter nano-objects, the effort has mainly been focused on the investigation of the “hot spot effect” of the nanoparticles’ surface and the thermometric performance has not been investigated, except in one recent work involving Fe_3_O_4_ functionalized by fluorescent polymer with Rhodamine B [31]. Note also that for these multifunctional IONPs, their magnetothermal properties have been targeted and their photothermal heating has never been explored in combination with thermometry despite a significant interest in recent years in the excellent photothermal capacity of IONPs.

In this article, we report on the synthesis, magneto- and photothermal heating and luminescent thermometry of multifunctional IONPs enwrapped into amphiphilic copolymers bearing luminescent coumarin moiety. Previously, some of us demonstrated the successful coating of IONPs by a hydrophilic polymer of the polyoxazoline family [33]. The amphiphilic polymer coating has been chosen in this work because this kind of macromolecular architecture allows a good affinity with inorganic nanoparticles and ensures a dispersion of IONPs in water thanks to its hydrophilic block. Further, the polyoxazolines are very attractive for biomedical applications because they have excellent biocompatibility, cyto- and haemocompatibility, as well as a stealth behaviour. Finally, it is expected that they can exhibit fluorescence due to their aromatic constituents [34], while, to the best of our knowledge, there have been no studies conducted on the temperature-dependent fluorescence of polyoxazolines.

Polyoxazolines can be easily synthesized by cationic ring-opening polymerization (CROP). This chemistry is a versatile tool for direct access to hydrophilic polymers using commercial 2-R-2-oxazoline monomers, where R corresponds to methyl or ethyl substituents, while hydrophobic ones result from monomers bearing propyl, butyl or phenyl substituent [35]. Hence, we elaborated two amphiphilic block copolymers, PPhOx_27_-PMOx_57_ and PButOx_8_-PMOx_42_, in one sequential step. Moreover, this polymerization process permits various terminal functionalizations using functional initiators and/or terminating agents. Consequently, amphiphilic block copolymers bearing a terminal coumarin unit, Coum-C_11_-PPhOx_27_-PMOx_59_ and Coum-C_11_-PButOx_8_-PMOx_42_ have been successfully prepared, as previously reported [36]. The introduction of the coumarin moiety is motivated by the following properties: (*i*) its ability to photodimerize under UV irradiation at 365 nm and, therefore, reinforce the IONP coating and (*ii*) its temperature-dependent emission in the visible region, making it a fluorescent thermometer [37]. However, although some polymers bearing a coumarin unit have already been investigated for temperature measurement purposes [38,39,40,41], they have never been used for the temperature detection and the encapsulation of nano-heaters.

In this article, a series of amphiphilic block copolymers, Coum-C_11_-PPhOx_27_-PMOx_59_ and Coum-C_11_-PButOx_8_-PMOx_42_, covalently bonded to a luminescent and photodimerizable coumarin moiety and PPhOx_27_-PMOx_57_ copolymer were evaluated in their micellar form as luminescent thermometers. All of them present a temperature-dependent fluorescence with the maximal relative thermal sensitivities (S_rmax_) varying in the range 1.90–2.71%·°C^−1^ at 60 °C. Secondly, they were tested for the encapsulation of spherical IONPs of 26 nm. The as-obtained multifunctional nano-objects IONP@Coum-C_11_-PPhOx_27_-PMOx_59_ are well dispersible in water and able to provide macroscopic heating triggered either by an AMF with a specific absorption rate (SAR) value of 240 W g^−1^ or laser irradiation with a photothermal conversion efficiency η = 68%. Moreover, they exhibit temperature-dependent luminescence, offering the function of a luminescent thermometer, which operates in the 20–60 °C range in water displaying the S_rmax_ value of 1.53%·°C^−1^ at 60 °C.

## 2. Materials and Methods

All the chemicals were purchased commercially and used without further purification. Ferric hydroxide oxide (FeO(OH) hydrated, 30–50 mesh), oleic acid (90%, OA) and oleylamine (90%, OL) were purchased from Sigma-Aldrich (Steinheim, Germany) and n-docosane (99%) was purchased from Acros organic. 2-Methyl-2-oxazoline (MOx), 2-butyl-2-oxazoline (BuOx) and 2-phenyl-2-oxazoline (PhOx) were dried and distilled from CaH_2_ and stored under a dry nitrogen atmosphere (purchased from Acros organic, Geel, Belgium). Acetonitrile (ACN) was distilled before use and stored under dry nitrogen (purchased from Acros organic). 7-Hydroxy-4-methylcoumarin (97%), 11-bromoundecanol (98%), methyl p-toluenesulfonate (MeOTs) (98%), ether, chloroform, p-toluenesulfonyl chloride (TsCl, >99%), triethylamine, pyridine, piperidine, MgSO_4_ and potassium carbonate were used without further purification and purchased from Acros organic.

### 2.1. Syntheses

*Amphiphilic copolymers.* The copolymers Coum-C_11_-PPhOx_m_-PMOx_n_ and Coum-C_11_-PButOx_m_-PMOx_n_ were synthesized according to already published procedures [36]. PPhOx_27_-PMOx_57_ was produced by using the same protocol and MeOTs as a cationic ring-opening polymerization initiator.

*Formation of micelles with amphiphilic copolymers*. Micelles were prepared in glass haemolysis tubes according to a film rehydration process. After weighing out approximately 20 mg of polymer, 1 mL of chloroform was added to solubilize it. CHCl_3_ was chosen for solubilization as it is a good solvent for both blocks, but it is also the solvent used for nanoparticles’ synthesis (later use). The solution was then evaporated using a rotary evaporator at 40 °C (pressure 104 mbar) for 1 h to obtain a thin, transparent and uniform film on the tube wall. Once the solvent was evaporated, the polymer was dried under dynamic vacuum for 3 h 30 and then under static vacuum for an additional 30 min. The film was rehydrated with an ultrapure water to achieve a concentration of 5 mg mL^−1^. The tube was stirred for 30 min at 65 °C and then sonicated at 65 °C for 1 h. The resulting solution appears as a clear, slightly opalescent one. It should be protected from light for storage. The same protocol was followed for all the amphiphilic copolymers. The dimerization of the micelles was achieved by irradiating the sample in a cylindrical photochemical “Rayonet” reactor. It is equipped with 16 symmetrical lamps emitting at 350 nm for dimerization and 254 nm for de-dimerization.

*Synthesis of pristine spherical IONPs stabilized by oleate and oleyl amine.* Pristine IONPs of ca. 26 nm stabilized by oleate (OA) and by oleyl amine (OAm) were prepared by adapting a previously published thermal decomposition method (at 350 °C) by using FeO(OH) as the iron precursor in n-docosane as a solvent [42]. First, a flask containing a mixture of FeO(OH) (2.1 mmol, 0.186 g), oleic acid (10 mmol, 3.17 g) and n-docosane (5.02 g) was connected to a Schlenk line to remove moisture and oxygen for 30 min at room temperature under vacuum and magnetic stirring. Subsequently, the flask was heated to 350 °C under argon flow with a heating rate of 10 °C min^−1^. The solution was maintained at 350 °C for a further 90 min under stirring and argon flow. Then, the mixture was cooled down to 200 °C. When the temperature reached 200 °C, the system was opened to air and the temperature was maintained at 180 °C for further 90 min to realize the nanoparticles’ oxidation (from FeO to Fe_3_O_4_). After this period, the heating was stopped. When the temperature of the suspension reached 50 °C, cyclohexane (15 mL) was added. The obtained nanoparticles were washed twice by dispersing in diethyl ether, followed by precipitation with ethanol (1:1 *v*/*v*), and were then recovered using centrifugation (20,000 rpm, 10 min). Oleylamine (200 µL) was added to the collected material as additional stabilizer. The resultant oleate/oleylamine-coated IONP/OA/OAm nanoparticles were dispersed and stored in chloroform (15 mL) [43].

*Encapsulation of IONPs by amphiphilic copolymers*. The encapsulation of IONPs by amphiphilic copolymers follows a similar protocol to the formation of micelles alone, except that a balloon is used instead of glass haemolysis tubes. After weighing out the 20 mg of polymer, 1 mL of a 1 mg mL^−1^ solution of IONPs in chloroform is added. Evaporation and vacuum drying were carried out in the same way as for the micelles alone. After the vacuum ramp, ultrapure water is added. Due to the magnetism of the IONPs, there is no need for an initial magnetic stirring step. The container is directly sonicated at 65 °C for around 1 h, alternating with vortexing to help solubilize the film. The obtained IONP@Coum-C_11_-PPhOx_27_-PMOx_59_ nanoparticles were washed three times with water. The nanoparticles were separated from nanoparticle-free micelles by using magnetic separation with a NdFeB- magnet (1 T).

### 2.2. Physical Methods

For transmission electron microscopy (TEM), 5 μL of suspension was deposited on a carbon-coated 300 mesh grid for 1 min, blotted dry by touching with a filter paper and then placed on a 2% uranyl acetate solution drop. After 1 min, the excess stain was removed by touching the edge with a filter paper, and the grid was dried at room temperature for a few min and examined using a Jeol 1400 Plus Transmission Electron Microscope (JEOL, Akishima, Japan) operating at 100 kV accelerating voltage. Data were collected with a high-sensitivity sCMOS JEOL Matataki Flash camera (JEOL, Akishima, Japan). Hydrodynamic diameter and polydispersity index (PDI) were measured using a Zetasizer NanoZS apparatus (Malvern Panalytical B.V., Almelo, The Netherlands) equipped with a He-Ne laser (wavelength: 633 nm) at a temperature of 25 °C and a scattering angle of 173° for detection. Size measurements were performed in water. ICP-AES analysis was performed by using a Spectro Arcos ICP (AMETEK Materials Analysis, Mahwah, NJ, USA). The samples were digested in nitric and hydrochloric acids before being diluted to obtain 10 mL of a final solution in 4% of acid.

*Photoluminescence and thermometry measurements.* Emission and excitation spectra were performed at room temperature (298 K) in water, using an Edinburgh FLS-920 spectrofluorometer (Edinburgh Instruments Ltd., Kirkton Campus, UK). The excitation source was a 450 W Xe arc lamp. The spectra were corrected for the detection and optical spectral response of the spectrofluorometer. Photoluminescent measurements as a function of temperature (luminescent thermometry) were performed by using the temperature setup incorporated into the Edinburgh spectrofluorometer. Emission spectra were recorded in the temperature range from 20 to 60 °C. At each temperature step, a period of 2 min was given to allow the temperature to stabilize and then 1 emission spectrum was recorded with a dwell time of 0.2 s and a step of 1 nm.

*Magnetothermia.* Magnetothermal experiences were realized using an alternating current magnetic field generator (UltraFlex Power Technologies, Ronkonkoma, NY, USA) at 342 kHz. The generating magnetic field is 20 mT at a frequency of 342 kHz. The samples were in liquid state (colloidal solutions) and isolated with polystyrene. The temperature of the liquid was measured using an OPTRIS PI 450 thermal camera (Optris, Berlin, Germany) and an optical fibre.

*Photothermia.* Photothermal experiences were realized using a laser with a wavelength of 808 nm and a surface power of 2.58 W.cm^−2^. The samples were in liquid state (colloidal solutions of 600 µL) in a glass tube. The temperature of the liquid was measured using an optical fibre.

### 2.3. Simulations and Fitting

*Magnetothermal experiments.* Magnetothermal experiences were fitted using the heat equation with a thermal exchange parameter (*f*(*T*)) as previously reported [43]:$$c_{v}\frac{dT}{dt}=P-f(T)$$
where *c_v_* and *P* are the heat capacity in J/K and the heat power source in W, respectively. *P* is linked to the *SAR* by the following equation:$$SAR=\frac{P}{m_{Fe}}$$
where m_Fe_ is the total mass of iron in the experiment sample.

*Photothermal experiments.* Photothermal experiences were fitted using a model built using COMSOL software [44]. The geometry of the experiments was reproduced in the software. The heat equation and the natural convection were solved simultaneously. The natural convection was solved for the nanoparticle suspension and for the air above the liquid. The heat equation was solved for the nanoparticle suspension, the air above and the glass tube. A thermal flux exchange was added between the glass and the rest of the environment. The thermal power source of the heat equation was calculated using the laser intensity and the shape of it spot, the light coefficient absorption *a* of the nanoparticle suspension and the photothermal efficiency *η*. The three following parameters were optimized during the fit process: the absorption, the thermal flux exchange and the thermal efficiency. The optimization of the parameters was realized using the least squares method in a python script which controls the COMSOL software.

## 3. Results and Discussion

The synthesis of the multifunctional nano-objects containing IONPs encapsulated by amphiphilic block copolymers was performed in three steps as follows: (*i*) the synthesis of a series of amphiphilic copolymers able to form micelles and the investigation of their thermometric ability (Figure 1a), (*ii*) the synthesis of pristine spherical IONPs of ca. 26 nm stabilized by oleate and oleylamine and (*iii*) the encapsulation of IONPs by amphiphilic copolymers and the investigation of their ability to macroscopically heat their environment under external *stimuli* (light irradiation and AMF) and provide a temperature readout (Figure 1b).

### 3.1. Amphiphilic Polymeric Micelles as Luminescent Thermometers

A series of amphiphilic copolymers was synthesized and investigated, with some of them functionalized by a hydrophobic and fluorescent coumarin moiety at the end of their hydrophobic block. The primary objectives were twofold: firstly, to explore their potential utility as luminescent thermometers, and secondly, to assess their suitability as luminescent shells for encapsulating hydrophobic IONPs. Each copolymer comprises a hydrophilic block of PMOx_n_ (where n corresponds to the number of hydrophilic repetitive units), ensuring the water dispersibility of further micelles. It was covalently bonded to a hydrophobic block denoted as PPhOx_m_ or PButOx_m_ (where m represents the number of hydrophobic repetitive units), essential for encapsulating IONPs. Two copolymers were functionalized by a coumarin moiety because the latter is fluorescent and also expected to undergo dimerization upon UV irradiation, thereby enhancing the stability of the micelles formed. Consequently, copolymer micelles containing coumarin were investigated both before and after dimerization to understand their behaviour comprehensively. Schematical representations of all the synthesized copolymers—namely, Coum-C_11_-PPhOx_27_-PMOx_59_ (P1 before and P2 after dimerization), PPhOx_27_-PMOx_57_ (P3) and Coum-C_11_-PButOx_8_-PMOx_42_ (P4 before and P5 after dimerization)—are shown in Table 1.

The amphiphilic block copolymers were prepared via a CROP process in one pot by sequential addition of the monomers using tosylate initiators. The latter included either a photosensitive coumarin modified by a C_11_ aliphatic spacer (Coum-C_11_-OTs) for the synthesis of Coum-C_11_-PPhOx_27_-PMOx_59_ and Coum-C_11_-PButOx_8_-PMOx_42_ copolymers [36], or a methyl group (MeOTs) for the PPhOx_27_-PMOx_57_. The polymerization under microwaves of hydrophobic PhOx or ButOx monomers was performed in order to obtain the hydrophobic block. Subsequently, the terminal reactive oxazolinium species of this block served as a macroinitiator to synthesize the hydrophilic PMO_x_ block, yielding the amphiphilic polymers. Finally, the reaction was quenched with piperidine to convert the oxazolinium end group into an unreactive terminal amine. The composition and the molecular weight of the as-obtained copolymers were determined by ^1^H NMR spectroscopy (Appendix A, ESI) and GPC (Appendix A, ESI).

An amphiphilic block copolymer, Coum-C_11_-PPhOx_27_-PMOx_59_, featuring a hydrophobic block of PPhOx covalently linked to the luminescent and photodimerizable coumarin moiety, was synthesized first. It is worth noting that aromatic moieties typically contribute to the fluorescence properties of organic molecules. For instance, pyrene functionalized oxazolines have already been studied for their fluorescent properties [34]. However, to the best of our knowledge, there have been no investigations of the fluorescence of commercial oxazolines and especially of a PPhOx hydrophobic block within amphiphilic block copolymers of the oxazoline family. Therefore, Coum-C_11_-PPhOx_27_-PMOx_59_ could potentially combine the luminescent properties of PhOx and coumarin, along with the potential stabilization of the micelles induced by the dimerization of the latter. The formation of micelles was performed in water by solubilizing the copolymer in chloroform, followed by drying to form films, which were then rehydrated in water (Appendix A, ESI). The micelles of Coum-C_11_-PPhOx_27_-PMOx_59_ could be obtained in either their dimerized or non-dimerized states by using reversible light irradiation at 350 nm for dimerization (transformation of P1 to P2) and at 254 nm for de-dimerization (transformation of P2 to P1) (Appendix A, ESI). To characterize the formed micelles, DLS measurements were performed on both the non-dimerized P1 and dimerized P2 forms. A monomodal population with a hydrodynamic diameter of 57 nm and a narrow dispersity (PDI: 0.12) was observed. The correlation and distribution curves are shown on Appendix A. Notably, no significant difference in the size of the formed micelles was observed before or after dimerization. The Transmission Electronic Microscopy (TEM) image of Coum-C_11_-PPhOx_27_-PMOx_59_ in its dimerized form (P2) demonstrates the presence of homogeneous spherical micelles.

In order to investigate the photoluminescence and encapsulation ability of polyoxazoline copolymer without coumarin, a block copolymer of similar composition and molar mass, PPhOx_27_-PMOx_57_ (P3), was synthesized and its micelles were prepared in water as described above. The DLS measurements indicated a hydrodynamic diameter of 50 nm with a narrow dispersity (PDI: 0.15) for P3 micelles (Appendix A, ESI).

Finally, Coum-C_11_-PButOx_8_-PMOx_42_ copolymer containing coumarin moiety, but with non-luminescent PButOx block instead of PPhOx, was synthesized and solubilized in water following the same preparation to form micelles. In this case, thanks to the presence of coumarin, micelles can also be obtained in either their dimerized or non-dimerized states by using reversible light irradiation at 350 nm for dimerization (transforming of P4 to P5) and at 254 nm for de-dimerization (transforming of P5 to P4). The DLS measurements revealed the presence of two populations, while the TEM images indicated the formation of rather cylindrical/vermicular-shaped micelles (Appendix A, ESI).

The photoluminescence properties of the aqueous solutions of copolymer micelles P1–P5 were first investigated in water at room temperature. Figure 1a demonstrates their excitation spectra monitored at 383 nm. The coumarin-free copolymer PPhOx_27_-PMOx_57_ presents only one band in the excitation spectrum located at 323 nm. The two coumarin-containing copolymers, Coum-C_11_-PPhOx_27_-PMOx_59_ and Coum-C_11_-PButOx_8_-PMOx_42_, demonstrate the presence of two (at 297 or 270 and 351 nm) or one (at 336 nm) excitation bands depending on the non-dimerized or dimerized states of coumarin, respectively.

The emission spectra of all the copolymer micelles shown in Figure 1b indicate the occurrence of a bright luminescence at 383 or 384 nm. The emission spectrum of the coumarin-free sample P3 under an excitation at 323 nm presents the emission band at ca. 383 nm, which can be attributed to the phenyl oxazoline moiety (PhOx). The coumarin-containing copolymer Coum-C_11_-PButOx_8_-PMOx_42_ exhibits a coumarin-linked emission at 383/384 nm in the non-dimerized (under excitation at 351 for P4) or dimerized (under excitation at 336 nm for P5) states of coumarin. Thus, the observed emission in Coum-C_11_-PPhOx_27_-PMOx_59_ stems from both fluorescent species, PhOx and coumarin. The excitation and emission wavelengths are summarized in Table 1.

Second, the emission spectra of P1–P5 were investigated in the 20–60 °C temperature range in order to evaluate their potential to work as luminescent thermometers. It is noteworthy that prior studies have explored the temperature-dependent luminescence of certain polymers featuring a coumarin moiety. Notably, research on coumarin-bearing thermosensitive copolymers, based on NIPAM and oligo(ethylene glycol) methacrylate blocks, has utilized the fluorescence of coumarin for temperature detection during phase transitions [40]. Similarly, PEG and P(NIPAM)-based copolymers, incorporating the same fluorescent coumarin moiety, have been investigated for luminescent temperature detection within living cells, leveraging the fluorescence resonance energy transfer (FRET) phenomenon [41]. Additionally, reports exist on coumarin-functionalized poly(vinyl alcohol) demonstrating a linear temperature dependence of the coumarin-based emission, although temperature detection remains unexplored [39]. It is important to note, however, that the temperature-dependent fluorescence of both coumarin- and oxazolines-based copolymer micelles, as well as the determination of their thermometric parameters, have not yet been investigated.

The temperature-dependent emission spectra in the 20–60 °C temperature range are demonstrated in Figure 2 for Coum-C_11_-PPhOx_27_-PMOx_59_ before (P1) and after dimerization (P2) and in Appendix A (ESI) for the other copolymers, P3–P5. As expected, the emission intensity decreases as the temperature increases. This effect has previously been attributed to the fluorescence quenching due to the increased collision between molecules with temperature and the intersystem crossing [39]. Note that after the first heating/cooling cycle, the intensity at 20 °C perfectly coincides with that primarily obtained at 20 °C, which indicates that no photodegradation/modification of the molecules’ structure occurred during the irradiation/heating. The photoluminescent intensity (integrated area between 370 and 500 nm) shows a linear temperature dependence for all the copolymers except for P4, for which a polynomial function can be used (Figure 3, Appendix A, ESI). Note that the coumarin-free copolymer P3 presents a higher intensity than P4, which offers the possibility to use it as promising luminescent temperature probe. The presence of 27 phenyl moieties for P3 instead of only one coumarin for P4 probably explains this difference.

The sensitivity of the luminescent thermometer is defined by the absolute sensitivity *S_a_*, which can be expressed as follows:(1)$$S_{a}\left(T\right)=\left|\partial I\left(T\right)/\partial T\right|$$

In order to compare the sensing performance of these copolymers with other organic dye-based thermometers, the maximal relative thermal sensitivity (*S_rmax_*) was calculated [23]. The *S_r_* value refers to the relative variation rate of the thermometric parameter (*I_380_* in the present systems) per degree of temperature, expressed as:(2)$$S_{r}\left(T\right)=S_{a}\left(T\right)/I\left(T\right)\times 100\%$$

The temperature dependence of *S_r_* for P1–P5 indicates that all the *S_r_* values are maximal at 60 °C (Figure 3, Appendix A, ESI). All the calculated *S_rmax_* values are in the range 1.90–2.71%·°C^−1^ (Table 2), which is superior to 1%·°C^−1^, the value frequently considered as a threshold for good luminescent thermometers [24].

The thermal uncertainties for copolymers P1–P5 were calculated as the smallest temperature change that can be calculated as follows:(3)$$\delta T=\delta I/S_{a}\left(T\right)=\delta I/(I\left(T\right)\times S_{r}\left(T\right))\;$$

The obtained thermal uncertainty values in the range 0.14–0.87 are satisfactory (Table 2). It is important to note that in Equation (3), the *S_r_* value is in °C^−1^ and not in % °C^−1^.

Considering the above-described results, all the prepared copolymer micelles present a good potential as emissive thermometers and can be used for encapsulation of IONP.

### 3.2. IONP Coating with Amphiphilic Bloc Copolymers

The luminescent amphiphilic copolymers Coum-C_11_-PPhOx_27_-PMOx_59_, Coum-C_11_-PButOx_8_-PMOx_42_ and PPhOx_27_-PMOx_57_ were used for the coating of spherical IONPs of 26 nm in order to combine the thermometric function of copolymers with the magneto or photothermal heating of nanoparticles (Figure 1b). The synthesis of the pristine IONP/OA/OAm was carried out by the classical thermal decomposition method coupled with the oxidation of FeO to Fe_3_O_4_ at the end of the procedure [43]. This step promotes the conversion of most of the FeO phase to Fe_3_O_4_. Note that the obtained nanoparticles stabilized by oleate and oleyl amine are very well dispersible in organic solvents and absolutely not dispersible in water. The encapsulation of the as-obtained IONP/OA/OAm nanoparticles by amphiphilic copolymers was carried out by using a three-step procedure, which consists of the following steps: (*i*) the solubilization of the IONP/OA/OAm and corresponding amphiphilic copolymer in an appropriate solvent, (*ii*) the formation of a film by solvent evaporation and (*iii*) solubilization of the film in water (Figure 2). A first indication of the successful encapsulation of the nanoparticles by Coum-C_11_-PPhOx_27_-PMOx_59_ and PPhOx_27_-PMOx_57_ was the complete change in behaviour in water as a homogeneous dark solution that was stable over days with no sedimentation. For Coum-C_11_-PButOx_8_-PMOx_42_, the nanoparticles sedimented within a few minutes, which indicated that the encapsulation failed. The cylindrical morphology of these micelles probably explained the difficulties encountered for the encapsulation. After the encapsulation step, a three-time magnetic washing was performed in order to remove the encapsulated IONPs from the formed empty micelles, which do not contain IONPs. Among the tested copolymers, a successful result was obtained only with Coum-C_11_-PPhOx_27_-PMOx_59_. Indeed, the nanoparticles’ aggregation/sedimentation was observed after the third washing for the PPhOx_27_-PMOx_57_/IONP system, which indicates the total loss of copolymer during the washing procedure. On the contrary, IONP@Coum-C_11_-PPhOx_27_-PMOx_59_ forms a nice colloidal aqueous solution of brown colour indicative of IONP presence. The photographs of each step of encapsulation during the formation of IONP@Coum-C_11_-PPhOx_27_-PMOx_59_ nano-objects are given in Figure 2. Note that this difference in the behaviours of Coum-C_11_-PPhOx_27_-PMOx_59_ and PPhOx_27_-PMOx_57_ was not expected and highlights that the presence of the coumarin moiety is necessary for the IONPs’ encapsulation.

The IONP@Coum-C_11_-PPhOx_27_-PMOx_59_ nano-objects were first characterized by DLS measurements, which indicated that the size of the micelles encapsulating the nanoparticles significantly increased to reach a hydrodynamic diameter of 128 nm with a narrow size distribution (PDI: 0.132) (Appendix A, ESI) in comparison with that of 57 nm (PDI: 0.120) obtained for the copolymer micelles alone (Appendix A, ESI). Note that after several washing cycles, the dispersity of nanoparticles becomes much narrower (PDI: 0.132 vs. PDI: 0.246 for the first and the third washing cycles), while the hydrodynamic diameter slightly increases (128 nm vs. 110 nm). This observation is coherent with the removal of the empty copolymer micelles during the washing. To confirm the DLS results, TEM observations were performed for the pristine IONP/OA/OAm and IONP@Coum-C_11_-PPhOx_27_-PMOx_59_ nanoparticles (Figure 4). The IONP/OA/OAm nanoparticles are quasi-spherical in shape and present a size of 26 ± 2 nm (Appendix A, ESI). In TEM, the electron density is responsible for the contrast of the objects observed. Thus, in Figure 4a, the Fe_3_O_4_ nanoparticles appear dark due to their electronic density compared to that of the carbon support of the grid used. To obtain the TEM image depicted in Figure 4b, negative staining was employed. This consists of applying an electronically dense molecular contrast agent, which increases the image contrast by darkening the area in which it is located. The experimental protocol followed here (see Materials and Methods) makes it possible to visualize the electronically dense Fe_3_O_4_ nanoparticle within the micelles, which appear light in colour, the contrast agent used for the negative colouring being located around the micelles.

### 3.3. Magneto- and Photothermal Heating

The magneto- and photothermal capacities of the IONP@Coum-C_11_-PPhOx_27_-PMOx_59_ nanoparticles were evaluated by measuring the temperature elevation of the aqueous colloidal solutions under both an applied AMF and laser irradiation at 808 nm.

The magnetothermal properties were investigated by using a previously described home-assembled device with a 20 mT/342 kHz field [43]. Temperature elevations of ΔT = 10.4 °C were observed after 15 min of exposure for an iron concentration of 0.637 mg_Fe_.mL^−1^ (Figure 5a). Note that the P2 solution (free of IONPs) used as a reference here provides a temperature increase of only 1.7 °C in the same conditions. The estimation of the SAR value was performed by using a phenomenological model grounded in the Newton temperature law, incorporating a thermal exchange function (see ESI for details) [45,46]. Typically, SAR values are used to evaluate the magnetothermal performance of magnetic nanoparticles by quantifying the generated heat power under an applied AMF [18]. The energy exchange function developed with the second-order Taylor series provided the best fit for the time dependence of temperature elevation curve (Figure 5a red solid line), allowing extraction of a SAR value of 240 ± 3 W.g_Fe_^−1^. This value is in the range of previously published ones obtained for efficient spherical IONPs working in water [43].

The photothermal properties of the IONP@Coum-C_11_-PPhOx_27_-PMOx_59_ nanoparticles were investigated in aqueous colloidal solutions under an 808 nm laser irradiation with a power of 2.58 W cm^−2^. The temperature of the colloidal solutions was monitored by using both an optical fibre immersed in the solution and a thermal camera. An important temperature increase of ΔT = 13.0 and 10.3 °C was observed under irradiation after 8 min with concentrations of 0.637 and 0.478 mg_Fe_.mL^−1^, while no obvious heating effect was detected in the water without nanoparticles (ΔT = 0.8 °C) (Figure 5b). The photothermal conversion efficiency, η, for the photothermal experiments was calculated by fitting the ΔT vs. time curves for both concentrations using a model built in the COMSOL software [44]. The absorption, the thermal flux exchange and the thermal efficiency were optimized in the fitting processes. The fitting details are given in the Materials and Methods section, as well as in Appendix A (ESI). The obtained η value (for both concentrations) stands at 68 ± 3%, positioning it in the higher end of values previously reported for IONPs [47].

### 3.4. Luminesence and Luminescent Thermometry

The photoluminescence of the IONP@Coum-C_11_-PPhOx_27_-PMOx_59_ nanoparticles was investigated in water and compared with that of the copolymer micelles of Coum-C_11_-PPhOx_27_-PMOx_59_ in their non-dimerized (P1) and dimerized forms (P2). The room temperature emission spectrum of IONP@Coum-C_11_-PPhOx_27_-PMOx_59_ performed in water under excitation at 323 nm exhibits a broad band at 383 nm ascribed to a Coum-C_11_-PPhOx_27_-PMOx_59_-characteristic emission (Figure 6). This result indicates that despite the presence of a relatively strong absorption of IONPs in the UV region, after encapsulation, the IONP@Coum-C_11_-PPhOx_27_-PMOx_59_ nanoparticles preserved the copolymer-characteristic emission, confirming successful coating with IONPs. The excitation spectrum of IONP@Coum-C_11_-PPhOx_27_-PMOx_59_ displays a single band at 323 nm (Figure 6) observed before or after dimerization under UV light. In contrast, the IONP-free micelles of Coum-C_11_-PPhOx_27_-PMOx_59_ show two bands at 270 nm and 351 nm in its non-dimerized form (P1) and one band at 336 nm in its dimerized (P2) state (Figure 1). This effect could be explained by the modification of the coumarin’s conformation in Coum-C_11_-PPhOx_27_-PMOx_59_ during the IONP coating process. Indeed, considering the fact that PPhOx_27_-PMOx_57_ was not able to provide stable IONP coating, we concluded that the presence of coumarin provides additional stabilization for IONP encapsulation.

Secondly, luminescence was investigated in water in the 20–60 °C temperature range in order to demonstrate the potential of these nanoparticles for temperature sensing. Figure 7a shows the temperature-dependent spectra of IONP@Coum-C_11_-PPhOx_27_-PMOx_59_ under 323 nm excitation and Figure 7b demonstrates the temperature dependence of the main intensity (I_383_) taken as an integrated area in the 370–500 nm range. The obtained intensity is linearly dependent on temperature, which confirms the possibility to use it for temperature sensing. The temperature dependence of *S_r_* is shown in the insert in Figure 7b. The maximum on this curve (*S_rmax_*) estimated from the calibration data is equal to 1.53%·°C^−1^ at 60 °C, which places these nanoparticles among the luminescent thermometers with a high relative thermal sensitivity (~1%·°C^−1^) [23]. Moreover, this value is coherent with the better *S_rmax_* values obtained for nanoparticles containing luminescent thermometers based on organic dyes working in water [48]. The measured thermal uncertainty of 0.82 °C is satisfactory. The thermometric parameters for IONP@Coum-C_11_-PPhOx_27_-PMOx_59_ nanoparticles are gathered in Table 2.

## 4. Conclusions

In summary, in this work, we reported on the synthesis and investigation of new multifunctional heater@thermometer nano-objects obtained by coating magnetic IONPs with luminescent coumarin-bound amphiphilic copolymer and which combine magneto- or photothermal heating with luminescent thermometry. Firstly, amphiphilic coumarin and polyoxazoline-based block copolymers, Coum-C_11_-PPhOx_27_-PMOx_59_ (P1 and P2 in its non- and dimerized forms) and Coum-C_11_-PButOx_8_-PMOx_42_ (P4 and P5 in its non- and dimerized forms) copolymers, as well as a coumarin-free copolymer, PPhOx_27_-PMOx_57_, were synthesized and evaluated for their utility as luminescent thermometers. All of them present temperature-dependent bright luminescence at 383 nm. Their temperature-dependent luminescence is well pronounced and displays a linearity (for all except P4), making them attractive as luminescent thermometers. Their maximal relative sensitivity values (S_rmax_) in the range 1.92–2.71%·°C^−1^ (at 60 °C) are rather satisfactory and indicate that these copolymers can be used for efficient temperature measurements. Secondly, the capacity of these copolymers to encapsulate spherical IONPs of 26 nm was investigated. Successful encapsulation was obtained only in the case of Coum-C_11_-PPhOx_27_-PMOx_59_. The as-obtained IONP@Coum-C_11_-PPhOx_27_-PMOx_59_ nano-objects are well dispersible in water, while the pristine IONPs immediately aggregate in water. The TEM images demonstrate the presence of single IONP core enwrapped by a shell of copolymer. The presence of a copolymer shell is also proved by the presence of a copolymer-characteristic bright luminescence at 383 nm.

The obtained IONP@Coum-C_11_-PPhOx_27_-PMOx_59_ nano-objects are able to provide a macroscopic heating triggered by an AMF with satisfactory SAR values of 240 W.g^−1^ or by light irradiation with photothermal efficiency of 68%. On the other hand, they exhibit the temperature-dependent luminescence of a coumarin-based copolymer shell, offering the function of luminescent thermometer at 383 nm in the 20–60 °C range in water, displaying a maximal relative thermal sensitivity of 1.53%·°C^−1^ at 60 °C.

## Data Availability

Data are contained within the article and Appendix A.

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
