# Peer review of "Magnetic Iron Oxide Nanoparticles Coated by Coumarin-Bound Copolymer for Enhanced Magneto- and Photothermal Heating and Luminescent Thermometry"

_nanomaterials, 2024, doi:10.3390/nano14110906_

Round 1

Reviewer 1 Report

Comments and Suggestions for Authors

This paper is clearly written and organized and the scientific work presented is relevant. Still, I have a number of issues that I would like to invite the authors to comment:

- Coumarins are sensitive to oxygen. Considering this, how the authors account for the different levels of dissolved oxygen concentration in media at different temperatures?

- Also, considering the potential impact of oxygen on the stability of coumarins can the authors comment about the stability of this molecular probe and their use during extensive periods of time?

- Could you comment about the impact of other possible interferences? Impact of turbidity? Inner filter effects?

- Could you also comment about the reproducibility of results?

- If temperature space mapping is the target, could you combine the approach presented in this work with PIV (Particle Image Velocimetry)?

Author Response

Montpellier, 16th of May

Dear Editor,

We are submitting the revised version of our manuscript ID: nanomaterials-3011778 entitled: 

Magnetic Iron Oxide Nanoparticles Coated by Coumarin-bound Copolymer for Enhanced Magneto- and Photothermal Heating and Luminescent Thermometry

by Alexiane Féron, Sylvain Catrouillet, Saad Sene, Gautier Félix, Belkacem Tarek Benkhaled, Vincent Lapinte, Joulia Larionova, Yannick Guari

We have carefully taken into account the criticisms of the reviewers and we thank them for their precious suggestions. We have listed below in more details our responses (in bold) to the reviewers’ criticism. The modifications in the manuscript performed are given in yellow in the manuscript.

Reviewer: 1

This paper is clearly written and organized and the scientific work presented is relevant.

We express our gratitude to the reviewer for his/her thoughtful feedback and the time he/she invested in reviewing our article.

Still, I have a number of issues that I would like to invite the authors to comment:

- Coumarins are sensitive to oxygen. Considering this, how the authors account for the different levels of dissolved oxygen concentration in media at different temperatures?

In our study, we used a protective strategy using amphiphilic polymer micelles based on oxazoline in order to overcome the sensitivity of coumarins to oxygen. These micelles create a protective environment where the coumarin moiety is shielded within the hydrophobic core. This configuration effectively hinders the diffusion of oxygen molecules towards the excited coumarin chromophores, thus mitigating their sensitivity to oxygen. To validate the photostability of coumarin-based micelles and the iron oxide nanoparticles enwrapped by coumarin-based copolymers, we conducted temperature-dependent fluorescence experiments by using several heating/cooling cycles. Remarkably, these experiments yielded consistent and reproducible results. Notably, the fluorescence intensity remained the same across successive cycles, indicating the robustness of our protective strategy. This consistency was also observed when employing amphiphilic copolymers bearing coumarin to encapsulate iron oxide nanoparticles (IONP). These findings suggest that our approach effectively maintains stable fluorescence levels across varying temperatures, regardless of the presence of oxygen in colloidal solutions made by freshly distilled water. On the other hand, our manuscript is dedicated to the proof of concept regarding the use of luminescent amphiphilic copolymers for the encapsulation of IONP in order to design multifunctional nanoparticles containing heater/thermometer in the same nano-object. We did not especially address the question of the coumarin based amphiphilic copolymers in the presence of different concentrations of dissolved oxygen. This point could be the object of further investigations.

- Also, considering the potential impact of oxygen on the stability of coumarins can the authors comment about the stability of this molecular probe and their use during extensive periods of time?

As written for the answer to the previous question, the photothermal stability of the amphiphilic coumarin-based micelles in their dimerized and non-dimerized states, as well as the stability of IONP enwrapped by amphiphilic copolymer Coum-C11-PPhOx27-PMOx59 was proved by using the fluorescence measurements during several heating/cooling cycles. The fluorescence intensity remains the same for all samples indicating that the conjugation of coumarin with amphiphilic copolymers based on oxazoline is an interesting strategy to overcome the photobleaching in the presence of oxygen (see Figure 2 and Figure S7 for the fluorescence intensity after several cycles for amphiphilic copolymer micelles and Figure 7b for IONP@Coum-C11-PPhOx27-PMOx59). On the other hand, the study of different (increased) oxygen concentrations and their impact on the photobleaching is out of the scope of the present article and could be the object of further study.

- Could you comment about the impact of other possible interferences? Impact of turbidity? Inner filter effects?

To minimize the influence of turbidity on the fluorescence measurements, we utilized sample preparation aimed to avoid micelles/particles’ aggregation (see Scheme 2 and Figure S3, S4). Note that photographs of micellar solutions shown in Figures S3 and S4 (ESI) were changed to improve the quality in order to demonstrate that the solutions are limpid. The concentration of all samples was controlled to ensure that potential turbidity levels remained within an acceptable range for accurate fluorescence measurements. Regarding possible inner filter effects, we conducted control experiments to assess the extent of inner filter effects in our samples and employed correction algorithms where necessary to account for any deviations caused by light absorption or scattering. Furthermore, the excitation and emission wavelengths were carefully chosen to minimize overlap with potential absorbing or scattering species present in the nanoparticles.

- Could you also comment about the reproducibility of results?

As previously mentioned, the fluorescence experiments on our samples were performed during three successive heating/cooling cycles indicating that the intensity was not altered by photobleaching and confirming the reproducibility of results.

- If temperature space mapping is the target, could you combine the approach presented in this work with PIV (Particle Image Velocimetry)?

Yes, the combination of our approach with PIV can offer valuable insights into the spatial variation of temperature within the fluid flow field and offers a comprehensive understanding of the coupled dynamics between fluid flow and temperature distribution, enabling detailed studies in various applications such as thermal management, microfluidics, and biomedical engineering. We thank the referee for this suggestion.

Referee: 2
Magnetic hyperthermia is a new, already used in practice, method of treating cancer in hard-to-treat areas by introducing biocompatible non-toxic magnetic nanoparticles into the tumor, their subsequent heating by an external electromagnetic field, leading to an increase in the temperature of tumor cells to +42°C and their subsequent death. Traditional methods of cancer tumor treatment are chemotherapy and radiation therapy, but the attention of scientists and physicians is turned to the search for alternative methods that would be less dangerous to the patient's health. Thus, the method of hyperthermia is considered as an alternative to the treatment of the last stages of cancer or as a supplement to traditional treatment, in which individual organs or parts of the organ affected by the pathological process are exposed to high temperature. The therapeutic effect of hyperthermia is due to the difference in response to heat between the two types of tissue - healthy and tumor tissue. Unlike healthy tissue, the affected tissue has large intercellular spaces, high density of blood vessels and poorly developed lymph nodes. Heat exposure to such tissue leads to damage and further cell death (apoptosis), while healthy cells remain unaffected. The therapeutic effect of hyperthermia is limited to the temperature range from +38ºC to 46ºC. At a temperature of +38 ºC active proliferation of tumor cells is observed, at +39 ºC viability decreases, and at temperatures above +43 ºC their death is observed. Also with the help of hyperthermia it is possible to reduce the size of the tumor to operable states. In the currently used hyperthermia methods (general hyperthermia, local hyperthermia using radiofrequency and microwave radiation as well as ultrasound), the heating of the tumor also causes a significant increase in the temperature of adjacent healthy tissues. Possible undesirable consequences of hyperthermia treatment are overheating, blood clots, burns and cardiovascular disorders. To control the thermal regime, the temperature is measured using sensors inserted into the treatment area. Thus, the main disadvantages of currently used hyperthermia techniques are low selectivity of exposure, as well as invasive method of temperature control.

The authors have proposed an original non-invasive method of measuring the temperature of magnetic nanoparticles heated by an alternating magnetic field by coating them with a luminescent polymer that changes its luminescence depending on the temperature. The work is new and certainly relevant.

We express our gratitude to the reviewer for his/her thoughtful feedback, his/her good opinion and the time he/she invested in reviewing our article.

Did the authors take into account the Brezovitch criterion for the combination of heating parameters - amplitude and field frequency – of magnetic nanoparticles?

The Brezovich criterion stipulates that the magnetic field - frequency product applied to the nanoparticles should be equal to or less than 4.85 × 108 Am-1s-1 in order to be used for the human body. In our case the product value is 5.44 × 109 Am-1s-1, which is approximately ten times higher. However, the Brezovich criterion is not the only one determined for hyperthermia clinic treatment. The literature analysis indicates that the magnetic field - frequency threshold value, which can be used for the safe clinical hyperthermia experiences is on discussion. We can cite for instance the Hergt criterion where the magnetic field - frequency product threshold determined as 5 ×109 (https://doi.org/10.1016/j.jmmm.2006.10.1156), or B. Thiesen & A. Jordan’s criterion of 1.8 ×109 (https://doi.org/10.1080/02656730802104757). When considering these various criteria, it can be concluded that our study is situated within an acceptable range. Moreover, our article is not focused on the hyperthermia treatment in vivo. The aim of our article is to demonstrate that the design of multifunctional heater/thermometer nanoparticles is possible to achieve by using the IONP encapsulation by luminescent amphiphilic copolymers and demonstrate: (i) their heating potential by using both, alternating current magnetic field and laser irradiation, and (ii) their thermometry by luminescence.

What is the Curie temperature of the synthesized nano-objects?

The referee certainly mentioned the blocking temperature of our nanoparticles (not the Curie temperature, which is the critical temperature for the bulk ferro- or ferrimagnetic materials). The magnetic properties of the pristine IONP have been reported previously (see Nanoscale 2022, 15, 144). The blocking temperature has been determined from the Zero Field Cooled/Field Cooled Magnetization curves as the maximum on the ZFC curve and situated above 320 K.

How do the authors plan to limit heating?

This is the object of our article devoted to the synthesis of IONP coated with fluorescent copolymer able to provide the temperature readout during the magnetothermal or photothermal heating. For heating limitation, lower frequency/amplitude of the magnetic field in the case of magnetothermal or lower laser power in the case of photothermal heating can be used. It is also possible to reduce the nanoparticles’ concentration.

From what temperature is the difference presented in Fig. 5?

The starting temperature is 20 °C. This has been added to the label of Figure 5

The authors may be interested in a potentially alternative way to measure temperature by coating the nanoparticles with a magnetocaloric coating and a temperature-sensitive polymer that changes its aggregate state upon heating/cooling https: //www.sciencedirect.com/science/article/pii/S2666032621000065

In any case, it is probably worth briefly mentioning in the introduction the existence of methods other than fluorescence-based temperature measurement of magnetic nanoparticles inside the body.

We sincerely appreciate the referee's valuable reference, which we will certainly incorporate into our future articles. While we acknowledge that there are various alternatives to fluorescence for thermometry readout, we must consider the balance of our article's focus. Adding a complementary paragraph at this juncture may potentially dilute the central theme of our work, as our introduction is already quite extensive. However, we will ensure to explore and discuss these alternative methods in depth in future publications. Thank you for your understanding.

Yours Sincerely,

Dr. Sylvain Catrouillet

Pr. Joulia Larionova,

Dr. Yannick Guari

Reviewer 2 Report

Comments and Suggestions for Authors

Magnetic hyperthermia is a new, already used in practice, method of treating cancer in hard-to-treat areas by introducing biocompatible non-toxic magnetic nanoparticles into the tumor, their subsequent heating by an external electromagnetic field, leading to an increase in the temperature of tumor cells to +42°C and their subsequent death.
Traditional methods of cancer tumor treatment are chemotherapy and radiation therapy, but the attention of scientists and physicians is turned to the search for alternative methods that would be less dangerous to the patient's health. Thus, the method of hyperthermia is considered as an alternative to the treatment of the last stages of cancer or as a supplement to traditional treatment, in which individual organs or parts of the organ affected by the pathological process are exposed to high temperature.

 The therapeutic effect of hyperthermia is due to the difference in response to heat between the two types of tissue - healthy and tumor tissue. Unlike healthy tissue, the affected tissue has large intercellular spaces, high density of blood vessels and poorly developed lymph nodes. Heat exposure to such tissue leads to damage and further cell death (apoptosis), while healthy cells remain unaffected. The therapeutic effect of hyperthermia is limited to the temperature range from +38ºC to 46ºC. At a temperature of +38 ºC active proliferation of tumor cells is observed, at +39 ºC viability decreases, and at temperatures above +43 ºC their death is observed. Also with the help of hyperthermia it is possible to reduce the size of the tumor to operable states.

In the currently used hyperthermia methods (general hyperthermia, local hyperthermia using radiofrequency and microwave radiation as well as ultrasound), the heating of the tumor also causes a significant increase in the temperature of adjacent healthy tissues. Possible undesirable consequences of hyperthermia treatment are overheating, blood clots, burns and cardiovascular disorders. To control the thermal regime, the temperature is measured using sensors inserted into the treatment area. Thus, the main disadvantages of currently used hyperthermia techniques are low selectivity of exposure, as well as invasive method of temperature control.

The authors have proposed an original non-invasive method of measuring the temperature of magnetic nanoparticles heated by an alternating magnetic field by coating them with a luminescent polymer that changes its luminescence depending on the temperature.
The work is new and certainly relevant.
Did the authors take into account the Brezovitch criterion for the combination of heating parameters - amplitude and field frequency - of magnetic nanoparticles?

  What is the Curie temperature of the synthesized nano-objects? How do the authors plan to limit heating? From what temperature is the difference presented in Fig. 5?

The authors may be interested in a potentially alternative way to measure temperature by coating the nanoparticles with a magnetocaloric coating and a temperature-sensitive polymer that changes its aggregate state upon heating/cooling https://www.sciencedirect.com/science/article/pii/S2666032621000065

In any case, it is probably worth briefly mentioning in the introduction the existence of methods other than fluorescence-based temperature measurement of magnetic nanoparticles inside the body.

Author Response

(The authors gave the same response as above.)
